# Atorvastatin Ester Regulates Lipid Metabolism in Hyperlipidemia Rats via the PPAR-signaling Pathway and HMGCR Expression in the Liver

**DOI:** 10.3390/ijms222011107

**Published:** 2021-10-14

**Authors:** Nan Hu, Chunyun Chen, Jinhui Wang, Jian Huang, Dahong Yao, Chunli Li

**Affiliations:** 1Department of Traditional Chinese Medicine, Shenyang Pharmaceutical University, Shenyang 110016, China; leymedes@163.com; 2Department of Pharmacology, Shenyang Pharmaceutical University, Shenyang 110016, China; ChunYun_Chen@163.com; 3School of Pharmacy, Harbin Medical University, Harbin 150000, China; wangjinhui@hrbmu.edu.cn (J.W.); huangjian@hrbmu.edu.cn (J.H.); 4School of Pharmaceutical Sciences, Shenzhen Technology University, Shenzhen 518060, China; yaodahong@sztu.edu.cn

**Keywords:** atorvastatin ester, hyperlipidemia, RNA-sequencing, PPAR-signaling pathway, gene expression

## Abstract

Atorvastatin ester (Ate) is a structural trim of atorvastatin that can regulate hyperlipidemia. The purpose of this study was to evaluate the lipid-lowering effect of Ate. Male Sprague Dawley (SD) rats were fed a high-fat diet for seven months and used as a hyperlipidemia model. The lipid level and liver function of the hyperlipidemia rats were studied by the levels of TG, TC, LDL, HDL, ALT, and AST in serum after intragastric administration with different doses of Ate. HE staining was used to observe the pathological changes of the rat liver and gastrocnemius muscle. The lipid deposits in the liver of rats were observed by staining with ORO. The genes in the rat liver were sequenced by RNA-sequencing. The results of the RNA-sequencing were further examined by qRT-PCR and western blotting. Biochemical test results indicated that Ate could obviously improve the metabolic disorder and reduce both the ALT and AST levels in serum of the hyperlipidemia rats. Pathological results showed that Ate could improve HFD-induced lipid deposition and had no muscle toxicity. The RNA-sequencing results suggested that Ate affected liver lipid metabolism and cholesterol, metabolism in the hyperlipidemia-model rats may vary via the PPAR-signaling pathway. The western blotting and qRT-PCR results demonstrated the Ate-regulated lipid metabolism in the hyperlipidemia model through the PPAR-signaling pathway and HMGCR expression. In brief, Ate can significantly regulate the blood lipid level of the model rats, which may be achieved by regulating the PPAR-signaling pathway and HMGCR gene expression.

## 1. Introduction

Hyperlipidemia is a pathological state of metabolic disorder that is generally characterized by referring to the anomalous rise of TC, TG, and LDL-C in plasma, and the anomalous decrease in HDL [1]. It is reported that the causes of hyperlipidemia are varied, such as unhealthy eating habits, psychology, diabetes, pathology, and heredity [2,3,4]. According to the 2015 Global Disease Burden Research data, metabolic diseases have become a key factor caused by the high incidence rates and high mortality rates worldwide [5].

The treatment of hyperlipidemia is divided into diet treatment and drug treatment [6]. The diet treatment of hyperlipidemia should follow the *First or Second step diet* of the American Heart Association for patients [7]. In fact, a plant-based diet containing healthy complex carbohydrates can reduce the risk of cardiovascular disease. Recent reports continue to support the view that consuming less fat and refined carbohydrates, as well as increasing the intake of fruits and vegetables, low-fat dairy products, and whole grain foods can protect people from cardiovascular disease. However, diet alone often insufficiently decreases serum cholesterol and it is difficult for many patients to maintain [8]. Current anti-hyperlipidemic medicines mainly include statins, fibrates, bile acid sequestrants, nicotinic acid, and cholesterol absorption inhibitors [9]. By inhibiting 3-hydroxy-3-methyl-glutaryl-coenzyme A (HMG-CoA) reductase, statins are the most effective lipid-lowering drugs currently and also are known as one of the most successful drugs to reduce CVD at the same time [10,11]. They can effectively reduce the total plasma cholesterol and LDL [12]. At the same time, they can also reduce TG and increase HDL in plasma to a certain extent and become commonly anti-cholesterol drugs [13,14]. Since 1987, atorvastatin and the other five available statins have been prescribed to prevent adverse cardiovascular events and to reduce both the total cholesterol and LDL levels in blood [15]. Lipoproteins are conventionally defined by their density as such: very-low-density lipoprotein (VLDL), intermediate-density lipoprotein (IDL), low-density lipoprotein (LDL), and high-density lipoproteins (HDL) [16]. Cholesterol fractions such as LDL and HDL cholesterol are the most common measure biomarkers in clinical medicine. Observational studies have shown that there is an inverse correlation between LDL and HDL [17]. In the early stage of atherosclerosis, due to the increased intimal permeability caused by vascular endothelial cell damage or exudation, a few plasma LDL penetrates the lower vascular endothelium, becomes stagnant in the vascular wall, and is further oxidized to the oxidized LDL (OX-LDL). The oxidized LDL is eaten by macrophages and forms foam cells. Foam cells continuously aggregate and fuse to form an atherosclerotic lipid core [18,19,20]. On the bright side, HDL can reverse cholesterol transport, inhibit the Ox-LDL, prevent the blood lipids from entering the inner wall of blood vessels, actively remove cholesterol formed by macrophages in the inner wall of blood vessels, and prevent the formation of the atherosclerosis. Therefore, it is an established fact that reducing the plasma LDL concentration can reduce the cardiovascular disease [21]. Plasma HDL cholesterol was associated with a reduced risk of myocardial infarction [22]. Atorvastatin is one of the most prescribed statins drugs in the world [23]. Therefore, it is essential to understand the related effects of atorvastatin. Currently, there are many pharmacological and medical studies on atorvastatin. First, it is very certain that atorvastatin has made an outstanding contribution to human health, but the adverse reactions of atorvastatin in the clinical application cannot be ignored [10,24]. The major reason for statins’ discontinuation is statin-induced myotoxicity (SIM) [25], with a 7% to 29% incidence rate of myotoxic reactions in patients treated with statins [26]. In addition, a major drawback associated with atorvastatin therapy concerns elevated serum transaminase. Therefore, atorvastatin is included in the list of the potential hepatotoxic drugs [27]. Statins appear to possess similar side effect profiles, differing only in their maximum potency [10].

Ate, with a molecular weight of 630.7574, was selected as the tested drug in this experiment. It was obtained by the structural modification of atorvastatin and provided by Harbin Medical University (Harbin, Heilongjiang, China). Therefore, the purpose of this study was to evaluate the lipid-lowering effect of Ate by detecting ORO-staining as well as biochemical indicators of blood lipids in rats; to explore whether Ate would have the common side-effects of statins by HE-staining, and to investigate its potential mechanism. This study will provide more insights into the potential mechanism of Ate.

## 2. Results

### 2.1. Characterization of Ate

As shown in Figure 1, the molecular formula and molecular weight of Ate were C_37_H_42_FN_2_O_6_ and 630.7574, respectively. ^1^H-NMR (400 MHz, CDCl_3_), δ(ppm): 7.21–7.14 (9H, m), 7.06 (2H, d, J = 7.9 Hz), 7.02–6.96 (3H, m), 6.86 (1H, s), 4.33–4.22 (1H, m), 4.19–4.07 (2H, m), 3.97–3.90 (1H, m), 3.77–3.72 (2H, m), 3.70 (1H, brs), 3.64–3.46 (3H, m), 2.46–2.40 (2H, m), 1.70–1.60 (2H, m), 1.54 (6H, d, J = 7.2 Hz), 1.49–1.43 (1H, m), 1.29–1.18 (5H, m); ^13^C-NMR (100 MHz, CDCl_3_), δ(ppm): 173.2, 172.3, 165.0, 162.4 (d, J = 247 Hz), 141.6, 138.5, 134.7, 133.3 (d, J = 9 Hz), 130.6, 128.9, 128.8, 128.5, 126.7, 123.7, 122.0, 119.7, 115.5 (d, J = 20 Hz), 69.7, 69.0, 68.1, 66.7, 63.8, 41.9, 41.4, 41.3, 39.2, 26.2, 21.9, 21.8, and 15.1.

### 2.2. Ate Can Significantly Reduce the Body Weight and Liver Index of the Hyperlipidemia Rats

The body weight of the experimental animals in each group is shown in Figure 2. Compared with the NFD group, the HFD group had a significant increase in the body weight at all time points. Compared with the HFD group, the body weight in the Ato 4.0 mg/kg group and Ate 2.0 mg/kg group had no change at each time point. Compared with the HFD group, the body weight of the Ate 4.0 mg/kg group decreased significantly at the second and third week after administration of Ate, and the body weight of the Ate 8.0 mg/kg group decreased significantly at all time points.

The liver index of the experimental animals in each group is shown in Figure 3. After 4 weeks of administration, the liver index of the HFD group was significantly higher than that of the NFD group and the liver index of both the group and Ate-dose groups was significantly lower than that of the HFD group.

### 2.3. Biochemical Analysis

After administration, the TC, TG, HDL, and LDL levels in serum were as shown in Figure 4A–D. Compared with the NFD group, the TC and LDL levels in serum of the HFD group increased significantly. After Ato and Ate treatment, the TC and LDL levels in serum of the HFD groups were significantly reduced. Compared with the NFD group, the content of TG in serum of the HFD group increased and decreased significantly after 4 weeks of treatment with 4.0 mg/kg Ato. The TG level in the Ate 8.0 mg/kg group decreased significantly after 4 weeks of treatment. Additionally there was no significant change in the TG level of the Ate 2.0 mg/kg and Ate 4.0 mg/kg groups. Compared with the NFD group, the HDL level in serum of the HFD group decreased significantly. After 4 weeks of treatment, the HDL level in serum of the Ato and Ate different-doses groups increased significantly.

The levels of ALT and AST in serum are shown in Figure 4E,F. Compared with the NFD group, the ALT and AST level in serum of the HFD group increased significantly. After 4 weeks of treatment with Ate and Ato, the ALT and AST content in serum of all the treatment groups were significantly decreased.

### 2.4. Histopathological Analysis

As shown in Figure 5A, Ate could improve hepatic steatosis and protect the liver. In the NFD group, there were abundant cytoplasms in the perivascular hepatocytes without steatosis. The HFD group had fat vacuoles of different sizes around the blood vessels. Compared with the HFD group, the Ato and Ate group had a certain degree of recovery of the liver cell structure, fat degeneration, and fat vacuole reduction.

As presented in Figure 5B, Ate could improve the lipid deposition in the liver of the hyperlipidemia-model rats. There was no lipid deposition in the liver of rats in the NFD group and the liver of rats in the HFD group showed a large area of orange-red, with serious lipid deposition. Liver lipid deposition decreased in different degrees in each dose group of Ate.

As presented in Figure 5C, Ate did not cause muscle toxicity in the experimental rats. In terms of the cross-sectional staining of the gastrocnemius muscle in the NFD group, it could be observed that the skeletal muscle cells at the staining site were arranged regularly; the nucleus was flat and oval; and the sarcoplasma was in the form of red and white muscle fibers, as indicated by the green arrow. The Ate 2.0 mg/kg group and Ato 4.0 mg/kg group showed a few nuclear movements inward. Compared with the NFD group, the HFD group, Ate 4.0 mg/kg group, and Ate 8.0 mg/kg group had no abnormal changes.

### 2.5. RNA-Sequencing Analysis

The NFD, HFD, and Ate 4.0 mg/kg group had three samples in each group and about 20 million reads were measured for each sample. Q 20% and Q 30% of the sequencing data of each sample were more than 98% and no less than 94%. The quality of the sequencing data was high, which meet the conditions for further analysis.

#### 2.5.1. Number of DEGs

The number of DEGs are shown in Figure 6A–D. In comparing the gene expression profiles of the HFD group and NFD group, 422 genes were significantly changed in the HFD group, of which 183 genes were up-regulated and 239 genes were down-regulated. Compared with the HFD group, 299 genes were significantly changed in the Ate group, of which 166 genes were up-regulated and 133 genes were down-regulated. There were 72 common DEGs in the HFD group, NFD group, and Ate 4.0 mg/kg group.

#### 2.5.2. GO Enrichment and KEGG Enrichment Analysis

The results of the GO enrichment analysis of differential genes are shown in Figure 6E. The terms shown in the figure are the top ten in descending order of the Q-values in this type. DEGs were the most highly involved in the cholesterol biosynthetic process in the biological process, in the ATP-binding cassette (ABC) transporter complex in cellular processes, and in monooxygenase activity in biological regulation.

To further understand the function of Ate, we performed a KEGG pathway analysis. As shown in Figure 6F, DEGs were significantly enriched in the PPAR-signaling pathway, retinol metabolism, steroid hormone biosynthesis, glycolysis/gluconeogenesis, biosynthesis of unsaturated fatty acids, and in other metabolic pathways in the KEGG pathway, although most significantly enriched in the PPAR-signaling pathway. These genes were *Cyp4a2*, *Scd*, *Pparg*, *Cyp4a8*, *Pck2*, *Pck1*, *RGD1565355*, and *Fabp4*. The expression amounts of these eight genes in different groups are shown in Figure 6G. DEGs involved in PPAR-signaling pathway construction through the KEGG (https://www.kegg.jp/, accessed on 15 August 2021) website are shown in Figure 7. Genes of *Pparg* and *Pck1* were lowly expressed in the HFD group and highly expressed in both the NFD and Ate 4.0 mg/kg group. On the contrary, Genes of *Cyp4a2*, *Scd*, *Cyp4a8*, *Pck2*, *RGD1565355*, and *Fabp4* were highly expressed in the HFD group and lowly expressed in both the NFD and Ate 4.0 mg/kg group.

### 2.6. Real-Time PCR Analysis

In order to further understand the RNA-sequence analysis, the genes related to the PPAR-signaling pathway with significant changes were selected from the suggested results and PPARα, PPARγ, CD36, HMGCS1, and LPL were verified by qRT-PCR. The results are established in Figure 8A–E. After modeling, the mRNA expression levels of PPARα, PPARγ, LPL, HMGCS1, and CD36 genes decreased significantly. After administration of Ate, the mRNA expression levels of PPARα, PPARγ, and LPL increased significantly, while the mRNA expression levels of HMGCS1 and CD36 genes decreased significantly.

### 2.7. Western Blot

In order to further explore the mechanism of Ate on the hyperlipidemia-model rats from the protein level, western blotting was used to detect the protein levels.

As shown in Figure 9, compared with the NFD group, the PPARα protein and HMGCS1 protein of the hyperlipidemia rats induced by HFD decreased significantly, while the expression of the HMGCR protein increased significantly. After 4 weeks of treatment, the expression of the PPARα protein and HMGCS1 protein increased significantly in a dose-dependent manner. HMGCR protein expression decreased significantly. Ato also significantly increased the expression of the PPARα protein and decreased the expression of the HMGCR protein significantly.

As shown in Figure 9, compared with the NFD group, the expression of the LPL protein and CD36 protein of the hyperlipidemia rats induced by HFD decreased significantly. After 4 weeks of treatment, the expression of the PPARγ protein and LPL protein increased significantly in a dose-dependent manner. However, the expression of the CD36 protein decreased significantly. Ato also increased the expression of the PPARγ and LPL protein significantly, and decreased the expression of the CD36 protein significantly.

## 3. Discussion

TG is the most abundant lipid in the human body, which mainly exists in chylomicrons (CM) and in very low-density lipoprotein (VLDL) particles in blood. If the TG level in blood is too high, it becomes too easy to deposit on the vascular wall and result in both vascular blockage and atherosclerosis. TC is one of the important bases for clinical detection and diagnosis of atherosclerosis and coronary heart disease [28,29]. Abnormal TC levels are often a component of atherogenic dyslipidemia, which is associated with decreased levels of HDL cholesterol and increased levels of small dense LDL particles [30]. Thus, it can be seen that when the levels of TG and TC in vivo are abnormal, the levels of HDL and LDL usually change. Consequently, in the screening of hyperlipidemia animal models, we believed that TC and TG in the serum of experimental animals increased significantly, indicating that the hyperlipidemia model is successful. Thus, when we studied the mechanism of Ate, TC, TG, LDL, and HDL were selected as indicators to evaluate the level of the blood lipid.

HFD is one of the factors causing hyperlipidemia. Excess blood lipid in the body will lead to hyperlipidemia and hepatocyte steatosis, resulting in a series of diseases such as fatty liver and liver cirrhosis. When hepatocytes are damaged, ALT and AST in the cytoplasm will be released into blood and their content in serum will increase significantly [31]. In this research study, Ate could significantly reduce the levels of ALT and AST in the hyperlipidemia rats and could alleviate liver injury. Ate had no muscle toxicity, proved it could regulate blood lipid, inhibited liver lipid deposition, and improved liver injury by HE-staining. It should be noted that the effect of 4.0 mg/kg Ate and 8.0 mg/kg Ate on reducing lipid, improving liver injury, and improving liver lipid deposition was better than Ato.

Peroxisome proliferator-activated receptors (PPARs) are ligand-activated transcription factors and comprise three subtypes, including PPARα, PPARγ, and PPARβ/δ [32]. They are encoded by different genes and are involved in several physiological processes including the modulation of cellular differentiation, development, the metabolism of carbohydrates, lipids and proteins, and tumorigenesis [33]. Many exogenous peroxisome proliferators (PP) are agonists of PPAR, as evidenced by the fact that PPARγ and PPARα are respective molecular targets for the type 2 diabetes drug thiazolidinediones (TZDs) and for dyslipidemia drug fibrates. Activated PPAR binds to retinol X receptor to form heterodimer and then binds to the peroxisome proliferator response element (PPRE), which can activate some proteins or enzyme genes related to fat metabolism [34,35].

PPARα is a member of the PPARs family and mainly located in the liver, heart, skeletal muscle, kidney, and brown adipose tissue ricing with active oxidation fatty acid β [36]. PPARα regulates various metabolic processes in the liver mainly by activating nuclear receptor mRNA, thus regulating the expression of downstream lipid metabolism-related genes and proteins, especially the expression of fatty acid β oxidation-related genes and proteins [37,38], such as transmembrane transport of fatty acid transport-related genes LPL and adenosine triphosphate-binding cassette transporter A1 [39]. Studies have shown that PPARα agonists can significantly reduce blood lipid and fatty liver in rodents, and PPARα agonists also have stable hypolipidemic effects in human clinical diseases [40,41,42]. It was found that compared with the control group, the model group of the PPARα gene knockout mice had prominent lipid deposition in the liver and the gene expression of lipid catabolism was down-regulated [43]. PPARγ is mainly distributed in adipose tissue and expressed in various organs and tissues such as liver and lymphoid tissue. PPARγ, a critical role in lipid metabolism, can regulate LPL, fatty acid translocase (CD36) phosphoenolpyruvate carboxy kinase (PCK), and other genes involved in fatty acid release, decomposition, transport, and storage [44,45]. PPARγ can also affect the signal pathways mediated by signal transcription factors and activator protein-1; can inhibit the activation of these pathways; and can achieve the purpose of inhibiting the activation and transcription of target gene promoters. PPARγ regulated glucose and lipid metabolism by these genes containing the PPRE structure, including hexyl COA synthase, LPL, insulin receptor substrate-2, and tumor necrosis factor [46]. In this study, the mRNA and protein expression of PPARα in the hyperlipidemia-model rats induced by HFD were significantly reduced. The blood lipids were abnormal and the liver fat lesions were obvious. After 4 weeks treatment with Ate, the mRNA and protein expression of PPARα and PPARγ increased significantly, and the dyslipidemia and liver lesions were improved to a certain extent, suggesting that PPARα/γ abnormal expression can lead to change in fat metabolism and result in the imbalance of both the lipid metabolism and lipid deposition in the liver. Ate can enhance the expression of PPARα/γ and restore it to the normal level, suggesting that Ate plays a role in regulating lipid metabolism, may regulate the blood lipid level, and may alleviate the liver lipid deposition by promoting the expression of PPARα/γ.

Clinical research found that 50% of 95 patients with hyperlipidemia suffered from fatty liver and most of them were hyperlipidemia at the same time [47]. It is thought that high TG caused by any reason can lead to hepatic steatosis [48]. The liver is the largest and most important metabolic site of the body. TG metabolism and cholesterol metabolism are the main energy metabolism processes of the liver. As the rate-limiting enzyme of TG degradation, LPL is the key factor for plasma TG clearance and tissue-uptake of fatty acids. It can hydrolyze TG in CM and VLDL into fatty acids and monoglycerides to provide energy for tissue oxidation [49,50]. The lipids absorbed by the human body from food are exogenous lipids, which are absorbed into blood in the small intestine to form CM. The lipids synthesized in the liver are endogenous lipids, which form VLDL and transport in the blood. LPL can decompose the above lipids into the free fatty acids and glycerol for the oxidation and energy supply of extrahepatic tissues. Abnormal LPL will lead to the accumulation of VLDL and CM in the body’s blood, leading to hyperlipidemia. Studies have shown that the serum TG level of mice without LPL in the liver is abnormally elevated and the lipid metabolism is seriously disordered. In addition, LPL-mediated TG hydrolysis to produce fatty acids can significantly reduce the expression of the cholesterol transporter in macrophages and then reduce both cholesterol outflow and hypercholesterolemia [51]. In this study, LPL mRNA and protein expression in the hyperlipidemia-model rats induced by a high-fat diet were significantly increased and their expression were regulated by the PPAR-signaling pathway [52]. The results showed that Ate could significantly up-regulate PPARα and PPARγ. Additionally, the expression of mRNA and protein suggested that Ate could up-regulate the expression of LPL, promote the decomposition of TG, reduce excessive TG in blood, and reduce liver lipid deposition.

CD36, namely fatty acid transposase, also known as a class B scavenger receptor and platelet glycoprotein IV, was considered to relate to lipid metabolism in 1993 when it acquired two new functions as a macrophage receptor for ox-LDL and as an adipocyte receptor/transporter for long-chain free fatty acids [53,54]. When studying the induction mechanism of ox-LDL on CD36, it was found that ox-LDL activated PPARγ to promote the expression of CD36 [55]. However, LDL does not have this mechanism. Conversely, HDL can have PPARγ phosphorylation inhibit the expression of CD36 [56]. In brief, the expression of CD36 was highly controlled. Ate could reduce the mRNA and protein expression levels of CD36 as well as improve the level of HDL, regulate the liver lipid metabolism in the link of the lipid transport, reduce the hepatocyte-uptake of the free fatty acid, and reduce the liver lipid deposition. As for the simultaneous decrease of the CD36 protein level in the HFD group, we believed that this is consistent with the previous theory. As shown in Figure 4, the level of HDL in the HFD group was lower than that in the NFD group. At the same time, it could be noted that the level of PPARγ in the HFD group was higher than that in the NFD group. Many CD36 ligands are closely involved in signal transduction processes. Therefore, we hypothesize that the decreased expression of CD36 may be due to the increased expression of phosphorylated PPARγ in the existing data analysis. The specific mechanism needs to be further studied.

Hyperlipidemia involves the imbalance of the cholesterol level. The increase of total cholesterol in the body is one of the main reasons for hyperlipidemia. The steady state of the cholesterol metabolism depends on its synthesis, absorption, and excretion pathways. Cholesterol in the body is mainly distributed in adipose tissue, about 70~80% of cholesterol is synthesized by the liver, and the liver is the main place for endogenous cholesterol synthesis. Cholesterol synthesis is a complex process and both HMGCS and HMGCR are the two key rate-limiting enzymes. HMG-CoA is synthesized by acetyl CoA under the action of 3-hydroxy-3-methylglutaryl coenzyme A synthase (HMGCS). Then, under the action of HMGCR, mevalonate is generated, the squalene is generated, and finally the cholesterol is generated [57,58]. Subsequently, more than 25 enzymatic reactions are required to synthesize cholesterol [59]. It is not difficult to observe that the balance of cholesterol is not only maintained by the synthesis of cholesterol. Rate-limiting enzymes play a key role in the synthesis process but not all perform this function. Therefore, the lower expression of the HMGCS1 protein in the HFD group compared to the NFD group was not surprisingly. However, it has been reported that the expression of the HMGCS gene will decrease when feeding excessive cholesterol [60]. The decrease of HMGCS1 expression may also be due to the negative feedback regulation of the body. Ate reduced the expression of HMGCR significantly, which was consistent with that of the HMGCR inhibitor Ato, and the effect of 8.0 mg/kg Ate was better than Ato. Ate decreased the expression of HMGCS1 and HMGCR significantly. There were no HMGCR and HMGCS1 in the target genes suggested by the transcriptome. The main reason why we included these two genes in this study is because Ate is modified by the structure of Ato. Therefore, through validation, we found that Ate could effectively reduce blood lipid and the mechanism may be different from that of Ato.

## 4. Materials and Methods

### 4.1. Animals and Treatments

Male SD rats weighing 250–280 g were obtained from the animal Center of Shenyang Pharmaceutical University (Shenyang, China; License: SCXK, Liaoning, 2020-0001). The animals were fed with commercial pellets and had access to both water and libitum, and were allowed to adapt to the environment for one week prior to the experiments. The room temperature was set at 22 ± 2 °C and the room humidity was set at 50% ± 20%. The animals used in this research study were approved in accordance with the Animal Ethics Committee of Shenyang Pharmaceutical University and the regulations were consistent with the ethical requirements of laboratory animals in China.

Fifteen rats were fed with the normal maintenance diet for seven months as the NFD group and the other animals were fed with the high-fat diet for seven months. The nutritional composition of the high-fat feed provided by Beijing BioPike Co., Ltd. (Beijing, China) and the results are shown in Table 1 The nutritional composition of the normal maintenance feed was provided by Beijing BioPike Co., Ltd., as shown in Table 2. When rats were fed the high-fat diet for 7 months, the TG and TC levels of the rats in the high-fat diet group were significantly higher than that of the rats in the normal diet group. Therefore, we regarded rats fed the high-fat diet as the stable hyperlipidemia models. Then, the hyperlipidemia-model animals were randomly divided into 5 groups, which consisted of the Ato 4.0 mg/kg group, Ate 2.0 mg/kg group, Ate 4.0 mg/kg group, Ate 8.0 mg/kg group, and high-fat diet group (HFD). Additionally, distilled water was used as the administration solvent. The HFD and NFD group were gavaged with the same amount of distilled water once a day for 4 weeks. The Ato 4.0 mg/kg group, Ate 2.0 mg/kg group, Ate 4.0 mg/kg group, and Ate 8.0 mg/kg group were given the therapeutic drug Ate and the control drug Ato by gavage once a day for 4 weeks. The rats in the NFD group were fed with the normal maintenance diet and the rats in the other groups were fed with the high-fat diet during the 4 weeks. They were weighed once before the beginning of the experiment and once a week after the beginning of the experiment. The dosage was adjusted at any time according to the change of body weight.

### 4.2. The Synthesis of Atorvastatin Ester (Ate)

Concentrated sulfuric acid (30 μL) in dropwise manner at 0 °C was added to a stirred solution of atorvastatin (558 mg, 0.1 mmol) in dichloromethane (10 mL) and 2-ethoxyethanol (2 mL). The resulting reaction mixture was stirred at room temperature for 3 h. Then, 10% NaHCO_3_ aq. (10 mL) was added to quench the reaction. The resulting mixture was extracted with ethyl acetate. The organic phase was washed with saturated brine. The combined organic phase was dried over anhydrous Na_2_SO_4_ and the solvent was removed under vacuum. The residue was purified by flash chromatography to afford the product as white solid with a yield of 53%.

### 4.3. Liver Index

The rats were fasted within 12 h after the last administration but could drink water freely. The liver was rinsed in pre-cooled normal saline and sucked dry with filter paper. The liver index was calculated by the following equation:Liver index = liver weight/body weight × 100%(1)

### 4.4. Biochemical Analysis

The rats were fasted within 12 h after the last administration but could drink water freely. Blood was taken from the orbit and placed in a 1.5 mL EP tube for 2 h at room temperature. Under the condition of 4 °C, the centrifuge was rotated at 3000 rpm/min and centrifuged for 15 min. The upper serum was absorbed and sub-packed, and was then placed in a refrigerator at −80 °C for storage.

TG, TC, LDL, and HDL in serum were quantified using assay kits (Meimian, Yancheng, Jiangsu, China) according to the manufacturer’s instructions. ALT and AST in the serum were also quantified using assay kits (Elabscience, Wuhan, Hubei, China).

### 4.5. Histopathological Analysis

HE-staining was performed to evaluate the general morphology of the liver and gastrocnemius muscle. The isolated rat livers and gastrocnemius muscles were fixed with 4% paraformaldehyde for more than 24 h and the fixed tissue was dehydrated with a gradient with alcohol. After paraffin embedding the dehydrated tissue, the tissue was cut into sections 3 µm thick. Then, paraffin-embedded slices were dewaxed with xylene and washed with hematoxylin eosin gradient ethanol for 5 min. After dehydration with conventional ethanol, the slices were sealed. Finally, the samples were observed with an optical microscope with a magnification at 40.0.

### 4.6. RNA-sequencing Analysis

The liver tissues of rats in the NFD group, HFD group, and Ate 4.0 mg/kg group were placed in sterilized 50 mL EP bottles, and RNA hold (TransGen Biotech, Beijing, China) 10 times the volume of the submitted tissue was quickly added to ensure that the samples were fully infiltrated. After that, the samples were sent to BGI Co., Ltd., Wuhan, China, for testing, with three samples for each group.

The tested samples were comprehensively evaluated according to the detection standard of BGI gene-sequencing samples and the results were determined according to the quality standard of RNA-sequencing samples. After the test results of the submitted samples were qualified, the next step of the transcriptome gene expression measurement could be carried out. According to the test results, the DEGs in each group were statistically analyzed to clarify the changes of the transcription-level genes after administration.

This test was carried out on the DNBSEQ (Wuhan, China) platform. SOAPnuke (v 1.5.2) filtered low-quality original readings containing connectors and more than 5% unknown *n* bases to clear the readings. After mapping the clean readings to the reference using bowtie 2 (v 2.2.5), the differential gene expression level was calculated using RSEM (v 1.2.12) and expressed as transcripts per million mapped reads (TPM). It is considered that when the absolute value of |Log_2_FC| > 0.585 and *p*-adjust < 0.05, the mRNA-level expression of samples has a significant difference and is then both enriched and analyzed by GO and KEGG.

### 4.7. Real-Time PCR Analysis

The total RNA of livers was isolated through the Trizol extraction method. Then, RNA was reverse-transcribed into cDNA according to a reverse transcription kit (TaKaRa, Shiga, Japan). GAPDH was regarded as a house-keeping gene. Real-time PCR amplification and detection were then performed with an CFX96 Real-Time PCR Detection System (Bio-Rad, Hercules, CA, USA). The primers used in this experiment were designed from NCBI and synthesized by Suzhou jinweizhi Biotechnology Co., Ltd. The primer sequences for real-time qPCR are listed in Table 3.

### 4.8. Western Blot

We added 100 per 10 mg of tissue μ L RIPA (PMSF: NaF: Na_3_VO_4_: RIPA = 1:1:100; Beyotime Biotechnology, Shanghai, China). After the tissue in RIPA was cut, it was then fully lysed with an ultrasonic crusher. The protein concentration was measured according to the instructions of the BCA kit (Beyotime Biotechnology, Shanghai, China) and a sulfate-polyacrylamide gel electrophoresis (SDS-PAGE) of appropriate concentration was configured according to the molecular weight of the target protein to separate the target protein. The target protein was then transferred to a PVDF membrane. After the membrane was blocked with 5% (*w/v*) non-fat milk for 1 h at room temperature, it was incubated with the primary antibody (anti-CD36 antibody, anti-PPARα antibody, anti-PPARβ antibody, and anti-HMGCR antibody) purchased from Abcam (Cambridge, Massachusett, UK), the anti-HMGCS1 antibody purchased from Cell Signaling Technology, Inc. (Danvers, MA, USA), the anti-LPL antibody purchased from Novus Biologicals, Inc. (Littleton, Colorado, USA), and the anti-β-actin antibody purchased from Beyotime Biotechnology Co., Ltd. (Shanghai, China), at 1:1000 dilution, at 4 °C overnight. The secondary antibody consisted of antibodies conjugated with horseradish peroxidase (1:4000 dilution, incubation for 1 h). The protein bands were detected using an ECL kit (Beyotime Biotechnology, Shanghai, China) and visualized using a Bio-Rad image analysis system (Bio-Rad, Hercules, CA, USA).

### 4.9. Statistical Analysis

All results are presented as mean ± SEM, both one-way ANOVA and Tukey’s test were used for comparisons between groups, and *p* < 0.05 was taken as the standard of significant difference. All statistical analyses were performed using SPSS 21.0 software and Graphpad Prism 8.02.

## 5. Conclusions

In summary, Ate expressed an obvious lipid-regulating impact on the hyperlipidemia-model rats, which can alleviate the liver lipid deposition and protect the liver. Its lipid-regulating effect may be realized by affecting the liver lipid metabolism through the PPAR-signaling pathway and HMGCR expression. The above results provide data to support further research of Ate, which, as a chemical drug with a clear structure, is a promising treatment for hyperlipidemia and various diseases caused by hyperlipidemia.

## Figures and Tables

**Figure 1 ijms-22-11107-f001:**
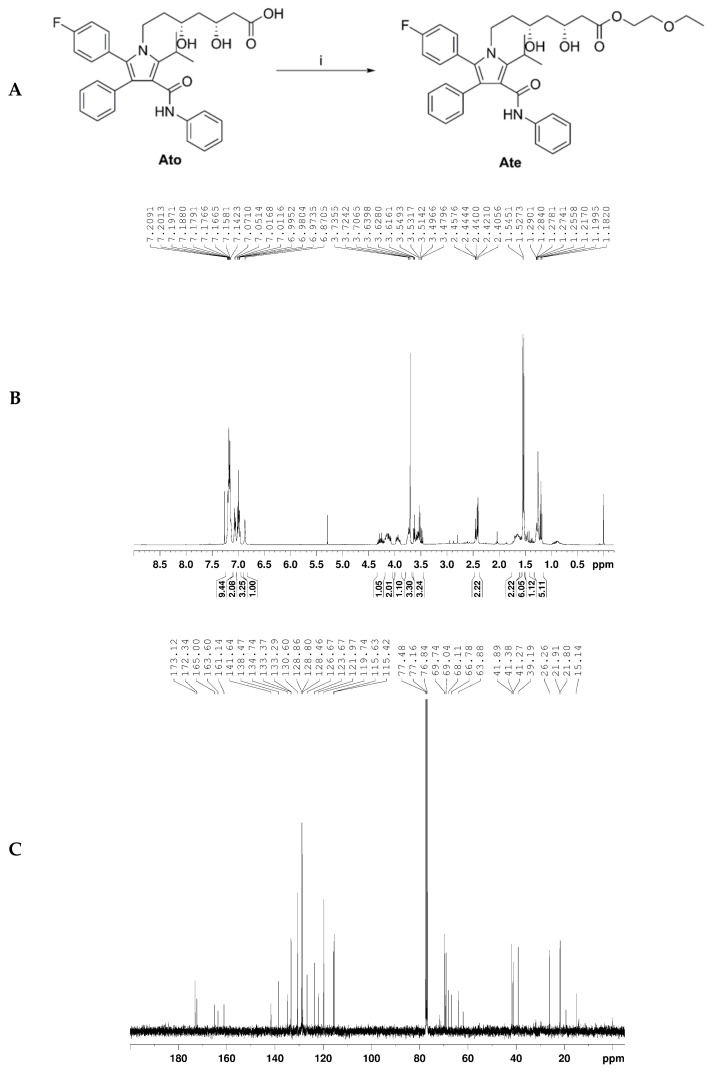
Characterization of Ate. (**A**) Ate was obtained after atorvastatin was modified. Reagents and conditions of Scheme i: 2-Ethoxyethanol, CH_2_Cl_2_, H_2_SO_4_, r. t., and 3 h. (**B**) The ^1^H NMR of Ate. (**C**) The ^13^C NMR of Ate.

**Figure 2 ijms-22-11107-f002:**
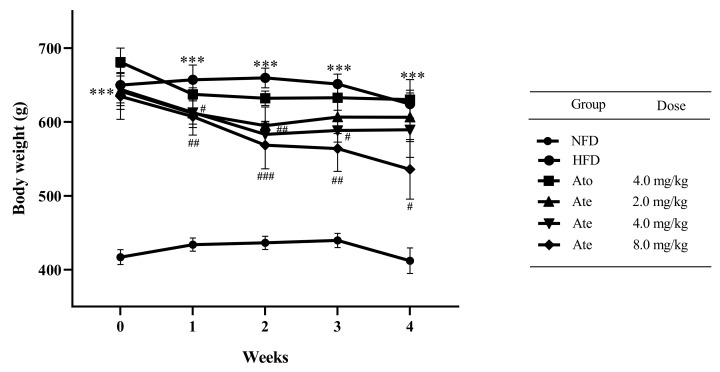
The effect of Ate on the body weight of rats. All data are shown as mean ± SEM, *n* = 15. *** *p* < 0.001 compared with the NFD group and # *p* < 0.05, ## *p* < 0.01, and ### *p* < 0.001 compared with the HFD group. The body weight of the HFD group, Ato 4.0 mg/kg group, Ate 2.0 mg/kg group, Ate 4.0 mg/kg group, and Ate 8.0 mg/kg group after being fed a high-fat diet for 7 months was regarded as the initial body weight of these five groups of rats. The body weight of NFD rats after being fed a normal diet for 7 months was regarded as the initial body weight of NFD rats.

**Figure 3 ijms-22-11107-f003:**
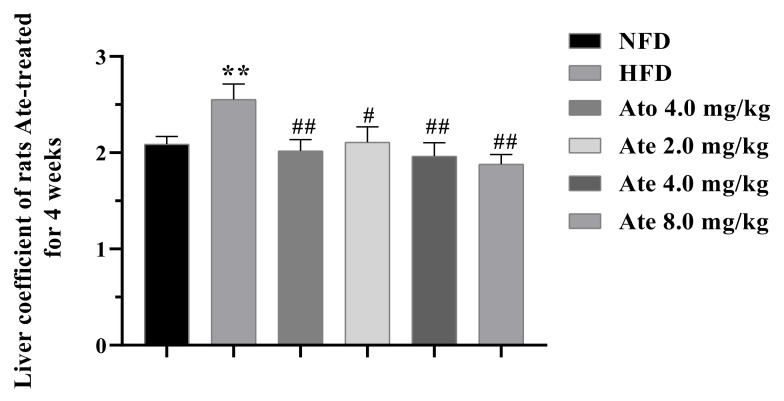
The effect of Ate on the liver coefficient of rats. All data are shown as mean ± SEM, *n* = 7. ** *p* < 0.01 compared with the NFD group and both # *p* < 0.05 and ## *p* < 0.01 compared with the HFD group.

**Figure 4 ijms-22-11107-f004:**
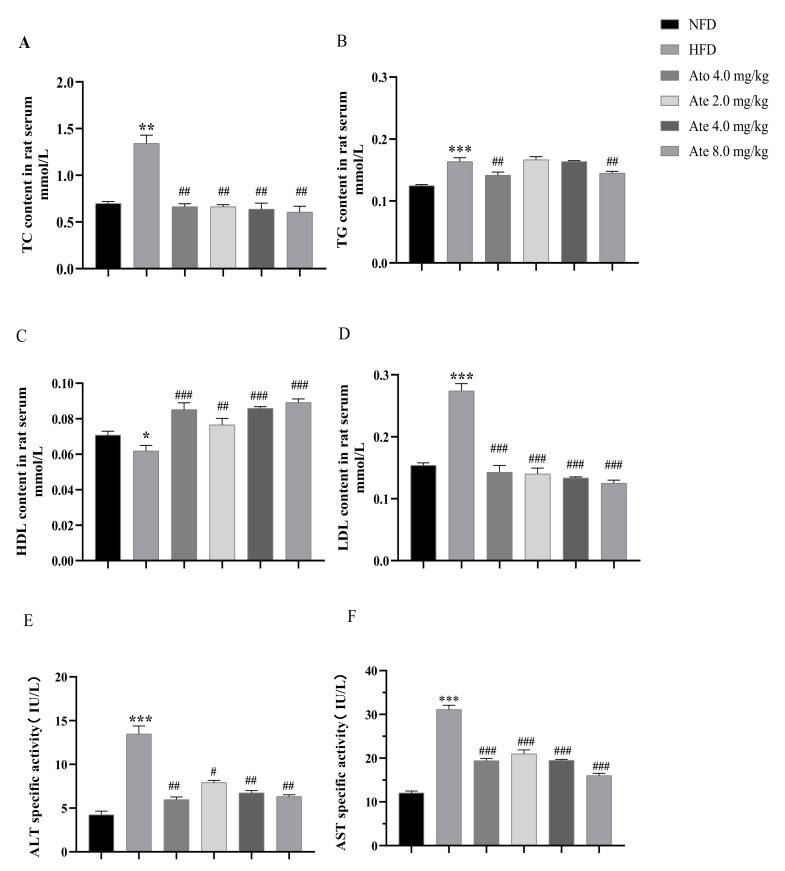
The effects of Ate on rat serum biochemistry. The levels of (**A**) TC, (**B**) TG, (**C**) HDL, (**D**) LDL, (**E**) ALT, and (**F**) AST in serum of rats are shown. All data are presented as mean ± SEM, *n* = 6–15. * *p* < 0.05, ** *p* < 0.01, and *** *p* < 0.001 compared with the NFD group; # *p* < 0.05, ## *p* < 0.01, and ### *p* < 0.001 compared with the HFD group.

**Figure 5 ijms-22-11107-f005:**
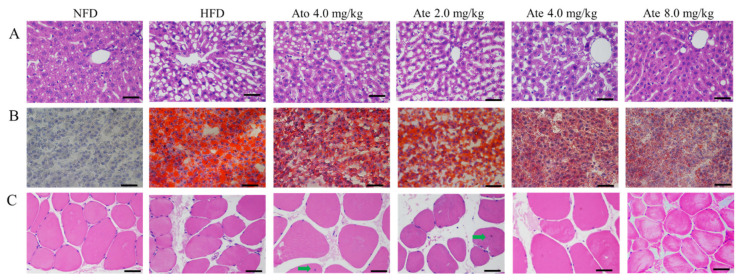
(**A**) HE-staining of rat liver in each group (400×): the cytoplasm of muscle cells is purple-red and the nucleus is blue. (**B**) Oil red O-staining of rat livers in each group (400×): the triglycerides were stained orange-red and the nucleus were stained blue. (**C**) Comparison of HE-staining in the cross-section of gastrocnemius muscle fibers in each group (400×): the cytoplasm of muscle cells is rose-red and the nucleus is blue. The green arrow indicates the inward migration of the nucleus.

**Figure 6 ijms-22-11107-f006:**
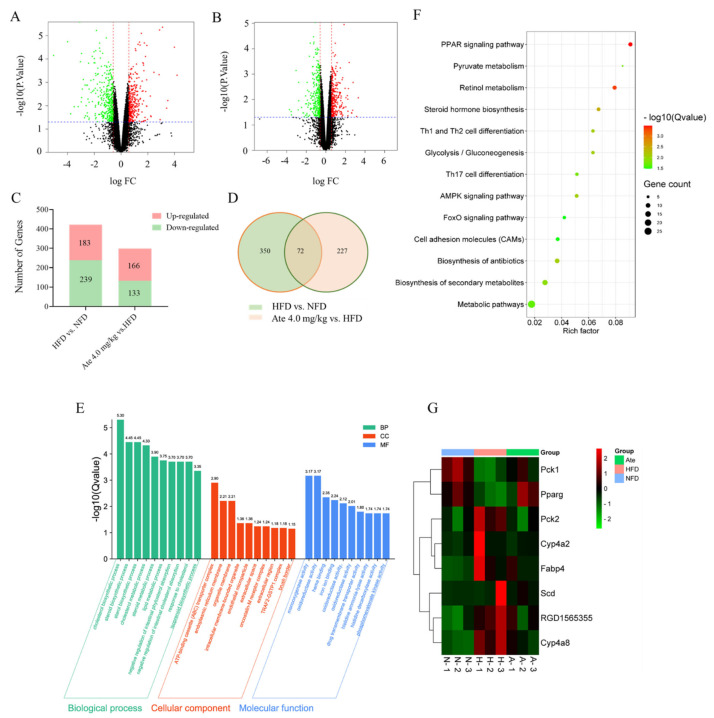
RNA-sequencing analyses of the livers’ DEGs in the NFD, HFD, and Ate 4.0 mg/kg groups. (**A**,**B**) Volcano diagram of DEGs in the livers between HFD group vs. NFD group and Ate 4.0 mg/kg group vs. HFD group. Red indicates up, green indicates down. (**C**) The numbers of up-regulated (red) and down-regulated (green) DEGs between the HFD group vs. NFD group and Ate 4.0 mg/kg group vs. HFD group. (**D**) Venn diagram of DEGs in between the HFD group vs. NFD group and HFD group vs. Ate 4.0 mg/kg group. (**E**) GO enrichment of those common selected DEGs including the biological process (green), cellular component (red), and molecular function (blue). We displayed the top 10 enrichment pathways of the Q-value in ascending order. (**F**) KEGG pathway enrichment histogram. All date values of the Q-value < 0.05. (**G**) Hierarchical clustering of the expression data for mRNA. Each column represents one sample; rows indicate the TPM of mRNAs. The relative expression of mRNAs is displayed according to the color scale. Green and red denote down-regulation and up-regulation, respectively. All date values of |Log_2_FC|> 0.585, *p* < 0.05, were accepted as significant.

**Figure 7 ijms-22-11107-f007:**
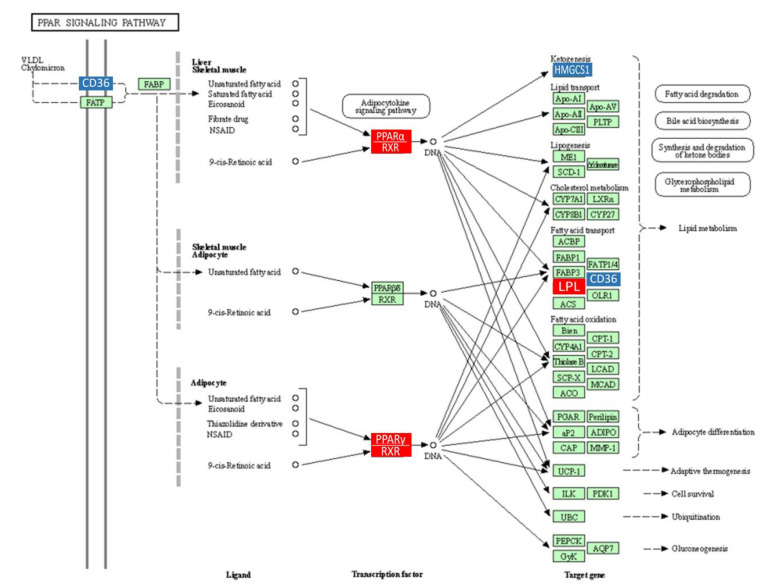
KEGG network diagram of the PPAR-signaling pathway. Red indicates up-regulated genes, blue indicates down-regulated genes, and green indicates genes that were not changed significantly.

**Figure 8 ijms-22-11107-f008:**
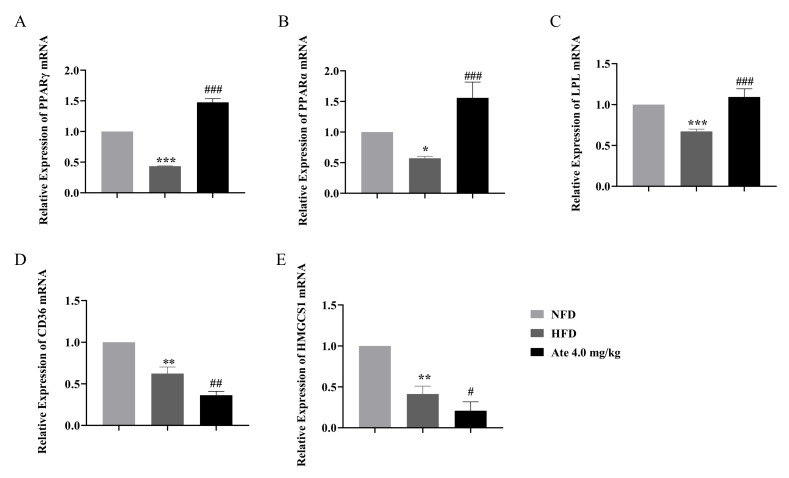
mRNA expression of genes related to the PPAR-signaling pathway. The gene expression of (**A**) PPARγ, (**B**) PPARα, (**C**) LPL, (**D**) CD36, and (**E**) HMGCS1 after modeling and treatment with Ate. All data are shown as mean ± SEM, *n* = 6. * *p* < 0.05, ** *p* < 0.01, and *** *p* < 0.001 compared with the NFD group; # *p* < 0.05, ## *p* < 0.01 and ### *p* < 0.001 compared with the HFD group.

**Figure 9 ijms-22-11107-f009:**
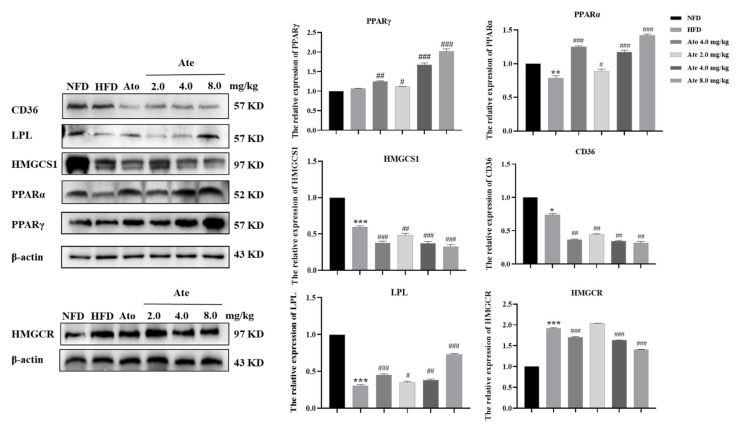
The effects of Ate on the PPAR-signaling pathway proteins and HMGCR protein expression in the liver of hyperlipidemia rats. The expression of PPARα, PPARγ, HMGCS1, CD36, LPL, and HMGCR after modeling and after treatment with Ate. All data are shown as mean ± SEM, *n* = 3. * *p* < 0.05, ** *p* < 0.01 and *** *p* < 0.001 compared with the NFD group; # *p* < 0.05, ## *p* < 0.01 and ### *p* < 0.001 compared with the HFD group.

**Table 1 ijms-22-11107-t001:** Composition of the high-fat feed formula.

Nutrient Composition	Per 1001.54 g (g)	Nutrient Reference Value
Casein, 80 mesh	195.00	19.47
DL Methionine	3.00	0.30%
Corn starch	50.00	4.99%
Maltodextrin	100.00	9.98%
Sucrose	341.00	34.05%
Cellulose	50.00	4.99%
Anhydrous milk fat	200.00	19.97%
Mineral mixture s10001	35.0	3.49%
Calcium carbonate	4.00	0.40%
Vitamin mixture v10001	10.00	1.00%
Choline tartrate	2.00	0.20%
Cholesterol	1.50	0.15%
Ethoxyquine	0.04	0.00%

**Table 2 ijms-22-11107-t002:** Composition of the normal maintenance feed formula.

Nutrient Composition	Per 1000 g (g)	Nutrient Reference Value
Water content	≤100	≤10%
Crude protein	≥180	≥18%
Crude fat	≥40	≥4%
Crude fiber	≤50	≤5%
Coarse ash	≤80	≤8%
Calcium	10–18	1–1.8%
Phosphorus	6–12	0.6–1.2%

**Table 3 ijms-22-11107-t003:** Primer sequences for quantitative real-time PCR.

Gene	Forward Primer (5′–3′)	Reverse Primer (5′–3′)
CD36	AACATCGAGTGTCGAATATGTGG	CCGAATAGTTCGCCGAAAGAA
HMGCS1	TGAACTGGGTCGAATCCAGC	CCTGTAGGTCTGGCATTTCCT
PPARα	AACATCGAGTGTCGAATATGTGG	CCGAATAGTTCGCCGAAAGAA
PPARγ	TCGCTGATGCACTGCCTATG	GAGAGGTCCACAGAGCTGATT
LPL	GGGAGTTTGGCTCCAGAGTTT	TGTGTCTTCAGGGGTCCTTAG
GAPDH	AGGTCGGTGTGAACGGATTTG	GGGGTCGTTGATGGCAACA

## Data Availability

Not applicable.

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
