# Peer review of "Atorvastatin Ester Regulates Lipid Metabolism in Hyperlipidemia Rats via the PPAR-signaling Pathway and HMGCR Expression in the Liver"

_ijms, 2021, doi:10.3390/ijms222011107_

Round 1

Reviewer 1 Report

The manuscript by Hu and colleagues investigates the use of a structurally modified statin, atorvastatin ester (Ate), in the treatment of hyperlipidemia using rats fed high-fat diet as a disease model. The authors have employed different concentrations of Ate and included a group of animals treated with a statin, for comparison. Overall, Ate had a protective effect against dyslipidemia, with no apparent liver toxicity. The protection could be associated with changes in the expression of genes relevant to lipid metabolism. The study is well designed and the conclusions are mostly supported by the data. However, the manuscript needs to be substantially improved for the sake of increasing clarity and readability. I enclose a list of issues that need to be addressed.

Major issues:

  1. The Abstract needs to state clearly what is the aim (or hypothesis) of the current work. For example, to investigate the effect of Ate treatment on hyperlipidemia in rats fed high-fat diet. Also, please include a final sentence summarizing the main conclusion(s) of the study.
  2. In the Methods section, please also provide the composition of the normal diet in Table 2, for comparison.
  3. P14, line 394: “When the TC and TG in serum of HFD group were higher than NFD group significantly, it was considered that the hyperlipidemia model was successfully established. Then the 395 hyperlipidemia model animals were randomLy divided into 5 groups…”. Please state for exactly how long the rats were fed HFD before starting the treatment with Ate or Ato.
  4. P4, lines 104-110 and Figure 2 legend: please state for how long the rats were fed HFD (or NFD) prior to treatment with Ate or Ato.
  5. The way the histological findings are reported needs improving: a) the images in Fig 5A show the presence of fat vacuoles in cytoplasm of hepatocytes surrounding a blood vessel, but it is not clear what the authors mean by “the structure of the hepatic lobules was not clear” or “the cytoplasm was loose”. Please document these observations with images or delete them. b) please correct the legend of Fig 5B because the images correspond to rat liver (not rat gastrocnemius). C) all the features reported in Figs 5A-C should be quantified or graded, otherwise authors should not refer to “significant” effects.
  6. The English grammar is very poor and some paragraphs are difficult to read. The authors should seek the help of a native English speaker or of a language editing service to improve the grammar. I enclose a few examples of corrections that should be made to make the article more readable, but a full list of corrections would be too extensive.

Minor issues:

  1. P1, line 15: replace “rats were feed high-fat diet” with “rats were fed high-fat diet”
  2. P1, line 33: replace “anomalously down of HDL” with “anomalous decrease in HDL”
  3. P1, lines 41-42: please replace “Recent reports continue to support the view that fat and refined carbohydrates” with “Recent reports continue to support the view that consuming less fat and refined carbohydrates” (or similar).
  4. P2, lines 52-54: the sentence does not read well. Delete “for 40 years”?
  5. P5, Figure 3: the tile of the y-axis is not clear (“of rats ato-treated”?).
  6. P6, line 137: a more appropriate title for Figure 4 would be “The effects of Ate on rat serum biochemistry”, because ALT and AST are not lipid-related parameters.
  7. P7, line 175: replace DGEs with DEGs.

Author Response

Thank you for your letter and comments for our manuscript entitled “Atorvastatin ester regulates lipid metabolism in hyperlipidemia rats via the PPAR signaling pathway and HMGCR expression in the liver” (IJMS-1397449). These comments are valuable and helpful in improving our paper and research. We have studied these comments carefully, and have made detailed responses following each comment. Revised portion are marked in red in the manuscript. The responses to the comment and actions taken are as following:

Responds to the reviewer’s comments:

Reviewer #1:

The manuscript by Hu and colleagues investigates the use of a structurally modified statin, atorvastatin ester (Ate), in the treatment of hyperlipidemia using rats fed high-fat diet as a disease model. The authors have employed different concentrations of Ate and included a group of animals treated with a statin, for comparison. Overall, Ate had a protective effect against dyslipidemia, with no apparent liver toxicity. The protection could be associated with changes in the expression of genes relevant to lipid metabolism. The study is well designed and the conclusions are mostly supported by the data. However, the manuscript needs to be substantially improved for the sake of increasing clarity and readability. I enclose a list of issues that need to be addressed.

Major issues:

  1. The Abstract needs to state clearly what is the aim (or hypothesis) of the current work. For example, to investigate the effect of Ate treatment on hyperlipidemia in rats fed high-fat diet. Also, please include a final sentence summarizing the main conclusion(s) of the study.

ResponseAction: Thank you for raising this question. The main conclusions that state the purpose of the current work and the content of the research are essential for the abstract. So, we have added the aim of this study in P1, line 14-15 and the conclusion in P1, line 27-28.

  1. In the Methods section, please also provide the composition of the normal diet in Table 2, for comparison.

ResponseAction: Thank you for raising this question and giving us such a good suggestion. In the revised manuscript, we have added the table 3 to show the nutrient composition of maintenance feed used in the normal diet.

Table 3. Composition of normal maintenance feed formula.

Nutrient composition

Per 1000 g (g)

Nutrient reference value

Water content

≤100

≤10%

Crude protein

≥180

≥18%

Crude fat

≥40

≥4%

Crude fiber

≤50

≤5%

Coarse ash

≤80

≤8%

Calcium

10-18

1%-1.8%

Phosphorus

6-12

0.6%-1.2%

  1. P14, line 394: “When the TC and TG in serum of HFD group were higher than NFD group significantly, it was considered that the hyperlipidemia model was successfully established. Then the 395 hyperlipidemia model animals were randomLy divided into 5 groups…”. Please state for exactly how long the rats were fed HFD before starting the treatment with Ate or Ato.

ResponseAction: Thank you for pointing out our omissions. When rats were fed the high-fat diet for 7 months, the TG and TC levels of rats in the high-fat diet group were significantly higher than rats in normal diet group. Therefore, we regarded rats fed the high-fat diet as the stable hyperlipidemia models. We explained this situation on page 14 line 390-393 of the revised manuscript. We have also corrected the spelling errors of “randomLy” in this paragraph.

  1. P4, lines 104-110 and Figure 2 legend: please state for how long the rats were fed HFD (or NFD) prior to treatment with Ate or Ato.

ResponseAction: Thank you for raising this question and giving us such a good suggestion. In the revised manuscript, we have added the time of rats were fed HFD (or NFD) before rats were treatment with Ate or Ato in Figure 2 legend.

  1. The way the histological findings are reported needs improving: a) the images in Fig 5A show the presence of fat vacuoles in cytoplasm of hepatocytes surrounding a blood vessel, but it is not clear what the authors mean by “the structure of the hepatic lobules was not clear” or “the cytoplasm was loose”. Please document these observations with images or delete them. b) please correct the legend of Fig 5B because the images correspond to rat liver (not rat gastrocnemius). C) all the features reported in Figs 5A-C should be quantified or graded, otherwise authors should not refer to “significant” effects.

ResponseAction: Thank you for your valuable advice. We must admit a misnomer in describing the pathological features of the tissue. We want to explain why these words are used, but those phenomena were described in Figure 5A. The reason is that we deleted the picture before submission. In the unchanged version, we still used the picture with lower magnification, so the hepatic lobule structure can be seen. After further consideration, we think Figure 5A is sufficient to illustrate the problem. Thank you very much for your careful review and help us find an important mistake. At the same time, we are very sorry for the inconvenience brought to your reading by our negligence. Therefore, we decided to delete the description that did not match the picture.

As you said, the tissue in this image is the liver of rats, not the gastrocnemius. We have corrected the legend in Figure 5B.

We did not quantify the features reported in Figure 5A-C and should not mention "significant" effects. Therefore, only the features in Figure 5A-C are objectively described

  1. The English grammar is very poor and some paragraphs are difficult to read. The authors should seek the help of a native English speaker or of a language editing service to improve the grammar. I enclose a few examples of corrections that should be made to make the article more readable, but a full list of corrections would be too extensive.

ResponseAction: Thank you for your careful review and professional comments for our manuscript! We feel sorry for the inconvenience in reading we have brought to you. Depending on your suggestions, we have made detailed and comprehensive improvement in English writing, including using more befitting words, more accurate grammar, and clearer expressions to make it more understandable and professional. Moreover, we invited English native speakers to help revise some sentences to make it more appropriate and natural. Some examples are as follows:

1) In line 64, 272 and 368, we corrected the wrong tenses we used;

3) In line 92, 96 and 468, we corrected the number that needed the lower corner;

4) In line 235,242 and so on,we have adjusted the word order. Make the sentence read more smoothly;

5) In line 81,103 and so on, we added “the” definite article "the" where we needed it;

Minor issues:

  1. P1, line 15: replace “rats were feed high-fat diet” with “rats were fed high-fat diet”

ResponseAction: Thank you so much for your suggestion. We have used the word “fed” instead of “feed” in line 15.

  1. P1, line 33: replace “anomalously down of HDL” with “anomalous decrease in HDL”

ResponseAction: Thank you very much for pointing out our mistakes. We have used the “anomalously down of HDL” instead of “anomalous decrease in HDL” in line 34.

  1. P1, lines 41-42: please replace “Recent reports continue to support the view that fat and refined carbohydrates” with “Recent reports continue to support the view that consuming less fat and refined carbohydrates” (or similar).

ResponseAction: Thank you very much for pointing out our mistakes. We have used the “Recent reports continue to support the view that consuming less fat and refined carbohydrates” instead of “Recent reports continue to support the view that fat and refined carbohydrates” in line 42-43.

  1. P2, lines 52-54: the sentence does not read well. Delete “for 40 years”?

ResponseAction: Thank you very much for your suggestions. We have deleted “for 40 years in line 53-55.

  1. P5, Figure 3: the tile of the y-axis is not clear (“of rats ato-treated”?).

ResponseAction: As you asked, we wrote the Y axis title wrong. We have corrected it in Figure 3 and thank you for your careful suggestion.

  1. P6, line 137: a more appropriate title for Figure 4 would be “The effects of Ate on rat serum biochemistry”, because ALT and AST are not lipid-related parameters.

ResponseAction: Thank you so much for your good evaluation and kind advice. According to your suggestion, we have changed the title of Figure 4 to “The Effects of Ate on Rat Serum Biochemistry”.

  1. P7, line 175: replace DGEs with DEGs.

ResponseAction: We are sorry for our oversight. We have replaced DGEs with DEGs in line 175.

Thank you very much for your careful suggestions again.

Reviewer 2 Report

First of all I want to congratulate the authors for this research.

Only some remarks I would like to make and please correct:

pag 2 of 19 row 58 please reformulate the phrase "Observational studies have shown that an inverse correlation between LDL and HDL" 

pag 3 of 19 row 88 in table 1 for ALT = Glutamate-pyruvate transaminase and not "Glutamate-private transaminase"

pag 11 of 19 row 273 "injuryby" (enter between injury and by)

pag 14 row 390-391 cut "with the" (is written duble times)

pag 14 row 396 correct randomLy in randomly.

pag 15 row 430 use HE (as it is in table 1) instead H&E 

Author Response

Thank you for your letter and comments for our manuscript entitled “Atorvastatin ester regulates lipid metabolism in hyperlipidemia rats via the PPAR signaling pathway and HMGCR expression in the liver” (IJMS-1397449). These comments are valuable and helpful in improving our paper and research. We have studied these comments carefully, and have made detailed responses following each comment. Revised portion are marked in red in the manuscript. The responses to the comment and actions taken are as following:

Reviewer #2:

First of all I want to congratulate the authors for this research.

Only some remarks I would like to make and please correct:

  1. pag 2 of 19 row 58 please reformulate the phrase "Observational studies have shown that an inverse correlation between LDL and HDL" 

ResponseAction: Thank you very much for pointing out our mistakes. We have used the “Observational studies have shown that there are an inverse correlation between LDL and HDL” instead of “Observational studies have shown that an inverse correlation between LDL and HDL” in line 42-43.

  1. pag 3 of 19 row 88 in table 1 for ALT = Glutamate-pyruvate transaminase and not "Glutamate-private transaminase"

ResponseAction: Thank you so much for your suggestions. We have used the word “Glutamate-pyruvate transaminase” instead of “Glutamate-private transaminase” in Table 1.

  1. pag 11 of 19 row 273 "injuryby" (enter between injury and by)

ResponseAction: Thank you so much for your suggestions. We have used the word “injury by” instead of “injuryby” in page 11 of 20 row 269

  1. pag 14 row 390-391 cut "with the" (is written duble times)

ResponseAction: Thank you very much for pointing out our mistakes. We have deleted the repetitive "with the" in page 14 row 384.

  1. pag 14 row 396 correct randomLy in randomly.

ResponseAction: Thank you very much for pointing out our mistakes. Spelling errors have been changed in page 14 row 394.

  1. pag 15 row 430 use HE (as it is in table 1) instead H&E

ResponseAction: Thank you so much for your suggestions. We have used the word “HE” instead of “H&E” in page 15 row 431.

Thank you very much for your careful suggestions again.
